**communications** engineering

# Inverted perovskite solar modules with 99.3% geometrical fill factor via nanosecond single laser patterning
Andrés E. R. Soto [●], Vera C. M. Duarte [●], Adélio Mendes [●] & Luísa Andrade [●] [✉]

Perovskite solar cells (PSCs) hold promise for high-efficiency photovoltaic technology but face commercialization challenges due to scaling difficulties. A common approach for scaling PSCs involves creating perovskite solar modules (PSMs) with subcells connected in series, using P1, P2, and P3 laser scribing process to reduce interconnection losses. In this study, a standard nanosecond pulse UV laser was used to perform these scribes. Here we demonstrated that, by employing a single 45 µm laser line for each scribe, it can significantly reduce the dead area, resulting in exceptionally high geometric fill factors (GFFs). In inverted PSMs with active areas of 4.0 cm$^2$ and 10.8 cm$^2$, it was reached GFFs of 99.3% and 98.8%, respectively. To the best of author's knowledge, this work demonstrates the first successful use of a single nanosecond laser source for continuous P1-P2-P3 scribing, achieving a dead area as low as 0.7% in a 4 cm$^2$ module.

Photovoltaic technologies (PVs) have been rapidly advancing in recent years, offering diverse options to generate clean energy from renewable sources, significantly contributing to the reduction of greenhouse gas emissions and the mitigation of climate change[1]. The scientific community has been working to enhance PVs, leading to the emergence of new technologies that are low-cost and use abundant earth materials[2]. Perovskite solar cells (PSCs) are part of this emergent family of PV technologies and have shown an extraordinary increase in power conversion efficiency (PCE) since the first reports, from 3.8% in 2009 to 26.7% in 2024[3,4], making them comparable to the market-leading PVs such as silicon solar cells[2].

PSCs are notable by their astonishing optoelectronic properties such as band gap tunability, long carrier lifetime, and high mobility, as well as advantages like semi-transparency, lightweight, color tunability, shape flexibility, and ease of fabrication[5,6]. PSCs can display two main configurations: the regular n-i-p and the inverted p-i-n (n = negative, i = intrinsic, and p = positive), which describe the order in which the active materials are layered in the device. The regular n-i-p configuration typically has the following structure: substrate/transparent conductive oxide (TCO)/electron transport layer/perovskite/hole transport layer /electrode, where the n layer is turned to the sunlight. The inverted structure has the same class of layers as the regular structure, but the charge carrier transport layers are inverted (Supplementary Fig. 1)[7]. Inverted PSCs usually demonstrate lower PCE than n-i-p perovskite structures due to the photovoltage loss resulting from the non-radiative recombination of photogenerated charge carriers[8]. However, this architecture is attracting increasing attention due to its simple fabrication processes without high-temperature sintering, negligible hysteresis

effects, and remarkable device operating stability[9]. A recent study reports an in-situ passivation methodology to enhance the perovskite layer, yielding superior photovoltaic performance due to the formation of well-oriented crystals and improved charge extraction at the hole transport layer/perovskite interface. This approach has led to a PCE of 26.7% for inverted devices, the highest value reported to date for a PSC[10].

Besides efficiency and stability enhancements, considerable research and innovation efforts are still required for PSC technology to establish a firm presence in the market, and the scaling-up challenges are undeniably considerable[2,11]. As the active area of the device increases, a decrease in efficiency can be expected due to the series resistance of the TCO electrodes, primarily caused by a reduction in the fill factor (FF)[12]. This challenge is critical in manufacturing perovskite PV modules, as the dimensions of a single subcell must be chosen so that the series resistance of the TCO electrodes does not affect module's performance. Moreover, inhomogeneous layer deposition in large-area solar cells also contributes to PCE loss. To address these challenges, three scribing steps are required to establish serial interconnections between the subcells of a module. These steps help to reduce the ohmic losses and to increase the output potential difference while maintaining a low current, similar to that in a small cell. This process, commonly referred to as the P1-P2-P3 scribing (Fig. 1), is currently the most widely used interconnection method across all PSC architectures and thin-film solar technologies[13,14].

Since PSCs are fabricated layer by layer using different deposition techniques[9,15], they can be subdivided either during or after their fabrication by masking or by mechanical or laser scribing, the latter being the preferable

LEPABE - Laboratory for Process Engineering, Environment, Biotechnology and Energy, ALiCE - Associate Laboratory in Chemical Engineering, Faculty of Engineering, University of Porto, Rua Dr. Roberto Frias, 4200-465 Porto, Portugal. [✉]e-mail: landrade@fe.up.pt

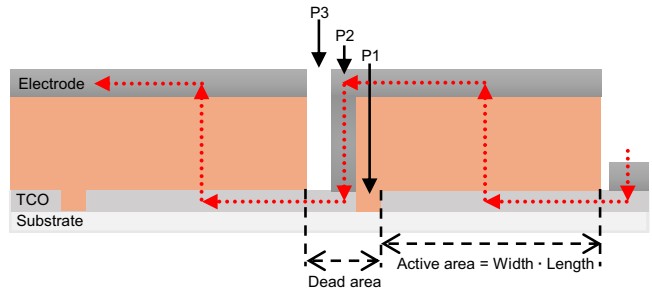

**Fig. 1 | Perovskite solar module showing areas defined by the P1-P2-P3 interconnection.** Red arrows indicate the direction of the current flow.

process for thin-film devices due to their versatility, precision, and smoothness[16,17]. Each P1-P2-P3 scribing step serves a function in interconnecting the subcells. P1 cuts the TCO on the substrate to pattern the subcell insulation. The P2 scribe removes the hole transport, perovskite, and electron transport layers to form each subcell, creating a low-resistance contact between the front and back electrodes. The P3 scribe removes all layers except the front electrode (TCO) to separate each subcell. This isolates the back electrode of neighboring cells and achieves a complete series connection module as shown in Fig. 1[18]. Perovskite solar modules (PSM) assembled as described above produce an approximate sum of the potential difference of each individual subcell while maintaining the current of the subcell with the lowest current value.

To assemble a functional PSM, two main areas must be optimized: the active area, which generates electrons for the external circuit, and the dead area, which serves as the connecting region between P1 and P3 scribes (see Fig. 1). Together, these areas constitute the aperture area. Measuring these areas allows the determination of the geometric fill factor (GFF), a crucial metric for assessing module interconnection, representing the ratio of active area to total aperture area. A higher GFF indicates reduced interconnection losses, a common problem in PSMs caused by charge losses within the inactive area and increased ohmic resistance in the interconnections[19].

Currently, there is extensive literature on laser ablation for processing regular PSMs, achieving high efficiencies with GFFs typically between 90% and 96%[14,16,20–24]. However, research on inverted PSMs designed with laser ablation lacks sufficient exploration of GFF and cell interconnection optimization (Table 1). Ritzer et al. demonstrated the highest GFF value of 96% for an inverted PSM with an active area of 3.84 cm² (PCE of 19.6%), employing conventional P1-P2-P3 interconnections with narrow scribes and two different nanosecond laser sources[25]. Rakocevic et al. achieved a GFF of 99% for an active area of 3.96 cm² (PCE of 10.4%) using a discontinuous point contact P2 scribing approach with a nanosecond laser, though this study focused on a regular PSC architecture[24]. Following a similar approach, Di Giacomo et al. recently reported a GFF of 99.6% for an inverted PSM with an active area of 2.61 cm² (PCE of 20.7%) using a picosecond laser and a discontinuous P2 scribe[26]. It is noteworthy that many studies on all-laser-scribed interconnections employ multiple laser sources, often with different wavelengths, and sometimes include picosecond and femtosecond lasers for precise ablation[24,26]. Shorter pulse durations generally offer finer control over ablation but come with increased complexity and cost, particularly for pulses shorter than 100 ps. Hence, nanosecond laser sources are favored for their practicality in industrial processes[24,26].

In this study, P1-P2-P3 interconnections were optimized for inverted PSMs using a conventional continuous pattern and leveraging a single nanosecond laser source for all scribes, compatible with industrial processes. Inverted PSMs with active areas of 4 cm² and 10.8 cm² were fabricated, achieving GFFs of 99.3% (PCE of 13.22%) and 98.8% (PCE of 9.57%), respectively. This work demonstrates the feasibility of continuous P1-P2-P3 interconnection with minimal dead area (0.7% for a 2-subcell module of 4 cm²) using a single nanosecond laser source for all three scribes.

## Methods
### Materials
The materials were used as received without further purification or treatment. Fluorine-doped Tin Oxide (FTO) glass substrates (TEC 7 Ω/□, Greatcell solar), Bathocuproine 99.99% (BCP, Sigma-Aldrich), Cesium iodide 99.9% (CsI, Honeywell), Formamidinium iodide (FAI, Greatcell solar), Lead (II) bromide 98% ($PbBr_2$, TCI), Lead (II) iodide 98% ($PbI_2$, TCI), Methylammonium bromide (MABr, Greatcell solar), [6,6]-Phenyl $C_{61}$ butyric acid methyl ester > 99% (PCBM, Ossila), Poly[bis(4-phenyl) (2,4,6-trimethylphenyl)amine] (MW = 13) (PTAA, Ossila), 2-(4-Fluorophenyl) ethylamine Hydroiodide (F-PEAI, TCI), Chlorobenzene anhydrous 99.8% (CB, Sigma-Aldrich), Isopropanol anhydrous (2-propanol, Sigma-Aldrich), Dimethyl sulfoxide anhydrous ≥99.9% (DMSO, Sigma-Aldrich), N,N-Dimethylformamide anhydrous 99.8% (DMF, Sigma-Aldrich), Toluene anhydrous (Sigma-Aldrich).

### Cell fabrication
The preparation of the inverted PSCs is based on the studies of Degani and colleagues[27]. All substrates were first patterned with a UV laser ablation system. Then, the substrates were gently brushed with a 10% Hellmanex aqueous solution and sonicated in an ultrasonic bath with distilled water twice for 5 min. Following, the substrates were cleaned with acetone and isopropanol for 10 min each in the ultrasonic bath. Finally, substrates were washed with absolute ethanol and dried with a nitrogen flow. Before the hole transport layer deposition, oxygen plasma was applied to the substrates for 10 min. All depositions were made inside a glovebox filled with nitrogen, except for the back contact. First, 50 μL of PTAA solution (1.5 mg mL⁻¹ dissolved in toluene) was spin-coated at a speed of 2000 rpm for 40 s, and then samples were annealed at 100 °C for 10 min. After annealing, the samples were treated with 100 μL of F-PEIA solution (5 mg mL⁻¹ dissolved in DMF) at 4000 rpm for 30 s. Next, the perovskite films were deposited. The perovskite precursor solution (1.2 M) is composed of a mixture of cations (Pb, Cs, FA, and MA) and halides (I and Br) dissolved in a solvent mixture of DMF/DMSO (4:1) according to a formula of $Cs_{0.05}(FA_{0.85}MA_{0.15})_{0.95}Pb(I_{0.85}Br_{0.15})_3$, with an excess of $PbI_2$ of 10%. The perovskite layer was fabricated using a two-step procedure: first, 60 μL of perovskite precursor solution was spin-coated at 1000 rpm for 12 s, and then at 5000 rpm for 27 s. At 21 s of the second spin-coating step, 100 μL of antisolvent (a 0.125 mg mL⁻¹ of F-PEIA solution on a mixture of 9:1 chlorobenzene/2-propanol) was dripped on the spinning substrate. Subsequently, the samples were annealed at 100 °C for 30 min. After the annealing process, substrates were cooled to room temperature. Then, 50 μL of PCBM solution (20 mg mL⁻¹ dissolved in chlorobenzene) was spun onto the perovskite layer at 2000 rpm for 30 s (at a ramp speed of 1000 rpm s⁻¹) and annealed at 100 °C for 10 min. Next, 50 μL of BCP solution (0.5 mg mL⁻¹ in 2-propanol) was spun at 4000 rpm for 30 s (at a ramp speed of 1000 rpm s⁻¹) as a buffer layer. Finally, the devices were masked and completed by thermally evaporating 80 nm of silver (Ag).

### Module fabrication
Modules were fabricated using a process similar to that of individual cells, with adjustments made to the amount of material before deposition. In addition, laser ablation was used to create three distinct scribes, referred to as P1, P2, and P3. P1 was performed before cleaning the substrate, P2 followed the deposition of all layers up to the BCP, and P3 was performed after the deposition of the silver back contact. All scribes were performed using a UV laser system equipped with a UV laser source (355 nm) of 5 W, a pulse frequency range of 20–200 kHz, and a minimum pulse width of 15 ns. A F-theta scan lens with a focal length of 160 mm was used, and an optimal working distance of 200 mm was determined.

The samples prepared for TLM analysis followed the same procedures as the module; however, the design was different, and no P3 was needed for TLM samples.

**Table 1 | Overview of the most relevant works published to date for inverted perovskite solar modules**

| Structure | Laser scribing conditions | PCE[a] (%) | Active area (cm²) | GFF (%) | Year/Ref. |
|---|---|---|---|---|---|
| Glass/FTO/NiO$_x$/CsPbI$_2$Br/ZnO-C$_{60}$/Ag | Not referred | 12.19 | 10.92 | 75.0 | 2020/[31] |
| Glass/ITO/PTAA/MA$_{0.7}$FA$_{0.3}$PbI$_3$/C$_{60}$/BCP/Cu | Not referred | 18.50 | 35.80[b] | 90.0 | 2021/[32] |
| Glass/ITO/PTAA/MA$_{0.7}$FA$_{0.3}$PbI$_3$/PFBS-C12/BCP/Cu | Not referred | 18.90 | 53.60[b] | 90.0 | 2022/[33] |
| Glass/ITO/NiO$_x$/Me-4PACz/Cs$_{0.1}$FA$_{0.9}$PbI$_{2.855}$Br$_{0.145}$/LiF/C$_{60}$/BCP/Cu | P1: 1064 nm, ps in N$_2$ P2, P3: 355 nm, ps in N$_2$ | 22.60 | 3.63 | 90.8 | 2023/[34] |
| Glass/ITO/NiO$_x$/Cs$_{0.05}$MA$_{0.14}$FA$_{0.81}$PbI$_{2.7}$Br$_{0.3}$/PCBM/BCP/Au | P1, P2, P3: 355 nm, ns | 15.90 | 10.20 | 90.9 | 2020/[18] |
| Glass/ITO/2-PACz/Cs$_{0.05}$MA$_{0.16}$FA$_{0.79}$PbIBr$_{0.51}$Cl$_{2.49}$/LiF/C$_{60}$/SnO$_2$/Cu | P1: 1064 nm, ps P2, P3: 532 nm, ps | 19.40 | 2.200 | 91.0 | 2021/[35] |
| Glass/ITO/BPT-1/Cs$_{0.17}$FA$_{0.83}$Pb(I$_{0.9}$Br$_{0.1}$)$_3$/C$_{60}$/BCP/Cu | P1, P2, P3: 355 nm, ns | 15.42 | 2.25 | 91.0 | 2022/[36] |
| Glass/FTO/Urea-NiO$_x$/Cs$_{0.05}$(MA$_{0.05}$FA$_{0.95}$)$_{0.95}$Pb(I$_{0.95}$Br$_{0.05}$)$_3$/BzMIMBr/C$_{60}$/BCP/Ag | P1, P2, P3: 532 nm | 17.18 | 178.4 | 91.0 | 2023/[37] |
| Glass/ITO/PTAA/Cs$_{0.08}$FA$_{0.92}$PbI$_3$/C$_{60}$/BCP/Cu | Not referred | 18.00 | 44.38[b] | 92.0 | 2021/[38] |
| Glass/ITO/PTAA/MA$_{0.6}$FA$_{0.4}$PbI$_3$/C$_{60}$/BCP/Cu | P1, P2, P3: 355 nm | 19.15 | 50.00[b] | 92.0 | 2021/[39] |
| Glass/ITO/PTAA/MAPbI$_3$/C$_{60}$/BCP/Cu | Not referred | 17.80 | 19.92 | 93.0 | 2020/[40] |
| Glass/ITO/NiO$_x$MAPbI$_3$/PCBM/BCP/Ag | P1: 1064 nm, ns P2, P3: 532 nm, ns | 15.70 | 19.16[b] | 93.0 | 2021/[41] |
| Glass/ITO/NiO$_x$-LS1/MAPbI$_3$/PCBM/BCP/Ag | P1: 1064 nm, ns P2, P3: 532 nm, ns | 14.90 | 19.16[b] | 93.0 | 2022/[42] |
| PEN/ITO/NiO$_x$/NP-NiO$_x$/SAM/perovskite/PEAI/C$_{60}$/BCP/Au | P1, P2, P3: 532 nm | 21.19 | 58.14 | 93.0 | 2023/[43] |
| Glass/ITO/PTAA/MAPbI$_3$/C$_{60}$/BCP/Cu | P1, P2, P3: 248 nm, ns | 15.00 | 57.20[b] | 93.4 | 2018/[44] |
| Glass/ITO/NiO$_x$/perovskite/PCBM/TBAOH/Ag | P1, P2, P3: 532 nm, ns | 7.20 | 46.00 | 93.8 | 2022/[45] |
| Glass/ITO/PTAA:BCP/Cs$_{0.1}$FA$_{0.9}$PbI$_3$/C$_{60}$/BCP/Cu | P1, P2, P3: 355 nm | 23.00 | 25.50 | 94.7 | 2023/[46] |
| Glass/ITO/PTAA/Al$_2$O$_3$/Cs$_{0.05}$(MA$_{0.1}$FA$_{0.9}$)$_{0.95}$Pb(I$_{0.9}$Br$_{0.1}$)$_3$/C$_{60}$/SnO$_2$/Ag | P1: 1064 nm, ns P2, P3: 532 nm, ns | 22.06 | 11.70 | 95.5 | 2022/[47] |
| Glass/ITO/SA-BPP/Cs$_{0.05}$(MA$_{0.22}$FA$_{0.78}$)$_{0.95}$PbI$_3$/PCBM/C$_{60}$/SnO$_2$/Ag | P2, P3: 532 nm, ps | 17.08 | 22.4[b] | 95.8 | 2023/[48] |
| Glass/ITO/spiro-TTB/MAPbI$_3$/C$_{60}$/BCP/Au | P1, P2, P3: 532 nm, ns | 19.6 | 3.84 | 96.0 | 2022/[25] |
| Glass/ITO/NiO$_x$/perovskite/LiF/C$_{60}$/BCP/Cu | P1, P2, P3: 355 nm, ps | 19.83 | 15.36 | 96.0 | 2023/[49] |
| Glass/FTO/PTAA/Cs$_{0.05}$(MA$_{0.15}$FA$_{0.85}$)$_{0.95}$Pb(I$_{0.85}$Br$_{0.15}$)$_3$/PCBM/BCP/Ag | P1:P2, P3: 355 nm, ns | 9.57 | 10.80 | 98.8 | This work |
| | | 13.22 | 4.00 | 99.3 | |
| Glass/ITO/PTAA/PFN-Br/Cs$_{0.05}$MA$_{0.14}$FA$_{0.81}$Pb(I$_{0.9}$Br$_{0.1}$)$_3$/PCBM/BCP/Cu | P1, P2, P3: 355 nm, ps | 20.70 | 2.61 | 99.6[c] | 2024/[26] |

[a]PCE based on the active area.
[b]Aperture area.
[c]Discontinuous interconnection.

## Characterizations

Photocurrent-voltage (J-V) characteristics curves of inverted PSCs were obtained by applying an external potential load and measuring the generated photocurrent using an Autolab (Metrohm Autolab, Netherlands) workstation controlled by the Nova software package (Nova 1.11). The measurements were made under 1 sun irradiation (AM 1.5, 100 mW cm$^{-2}$) supplied by a solar simulator (150 W Oriel class A). The simulator was calibrated using a single-crystal Si photodiode (Newport, USA). A scan rate of 20 mV s$^{-1}$ and a voltage step of 10 mV were used for individual cells, and a scan rate of 50 and 100 mV s$^{-1}$ and a voltage step of 10 mV were used for modules. The devices were characterized right after their preparation, at room temperature and in ambient air. The external quantum efficiency of modules was measured (Newport Corporation) to ensure a mismatch of the integrated and measured J$_{SC}$. Resistance measurement of TLM samples was performed with a Keithley 2425-C SourceMeter in 4-wire mode, using round gold probes. SEM-EDX characterization was performed using a desktop SEM Phenom XL.

## Results and discussion

Inverted perovskite solar cells have exceptional properties and advantages that position them as a leading choice in a diversified PV market serving various applications. While the device architecture mainly influences

module performance, it must also align with manufacturing conditions suitable for PSM fabrication. Kosasih et al. demonstrated that high-temperature annealing of charge transport layers during mesoporous n-i-p module fabrication allows sodium diffusion from the soda-lime glass substrate into P1 lines, thereby affecting module power output negatively[28]. Moreover, compatibility with flexible substrates is crucial. Adopting an inverted architecture mitigates these issues and ensures better compatibility with diverse manufacturing conditions and substrates.

After selecting the desired architecture, two criteria are essential to minimize upscaling losses: (1) achieving homogeneous and defect-free deposition of all functional layers and (2) fabricating module interconnections with optimal electrical properties and minimal lateral extension. To illustrate the first criterion, inverted PSCs of different active areas (0.2, 3, 5, and 10 cm²) were fabricated using a glass/FTO/PTAA/F-PEIA/Cs$_{0.05}$(FA$_{0.85}$MA$_{0.15}$)$_{0.95}$Pb(I$_{0.85}$Br$_{0.15}$)$_3$/F-PEIA/PCBM/BCP/Ag structure. The F-PEIA interface enhances the power conversion efficiency and device stability by improving perovskite film quality[27]. Photovoltaic parameters are detailed in Fig. 2 and Supplementary Fig. 2, showing that the 0.2 cm² cells primarily achieved average power conversion efficiencies of 16%. Notably, a maximum power conversion efficiency of 18.80% was achieved. Comparing the best-performing devices, scaling from 0.2 to 3 cm² resulted in a 15-fold area increase with a 22% decrease in power conversion efficiency. However,

**Fig. 2 | Photovoltaic performance of inverted PSCs for different active areas. a** J-V curves of inverted PSC champion device for each size (The arrow indicates increasing active area; solid lines represent forward scan and dashed lines reverse scan); **b** box plot of power conversion efficiency evolution with active area; **c** photovoltaic performance parameters of inverted PSC champion device for each size.

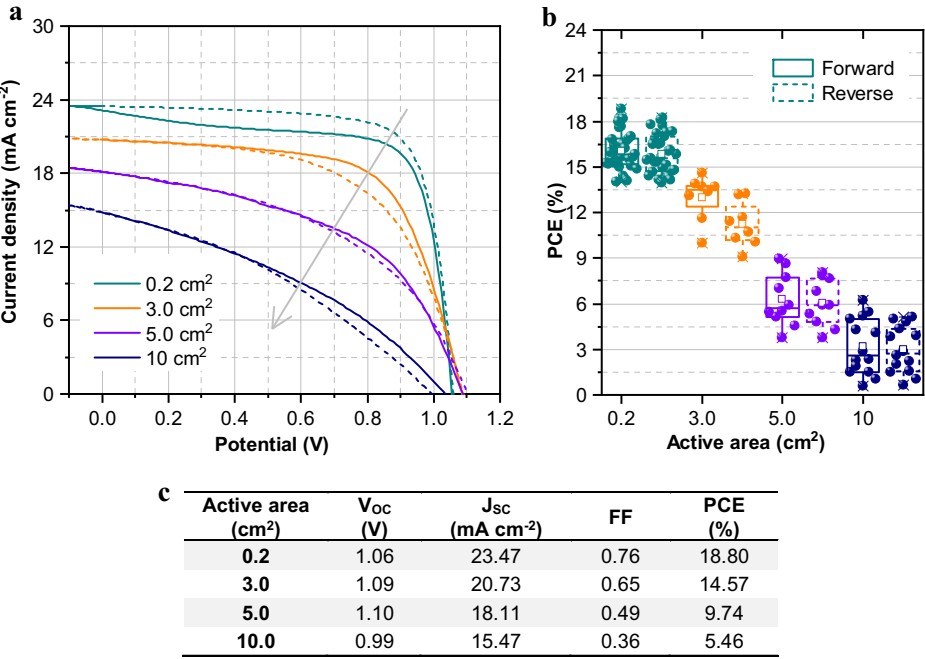

| Active area (cm²) | V_OC (V) | J_SC (mA cm⁻²) | FF | PCE (%) |
|---|---|---|---|---|
| **0.2** | 1.06 | 23.47 | 0.76 | 18.80 |
| **3.0** | 1.09 | 20.73 | 0.65 | 14.57 |
| **5.0** | 1.10 | 18.11 | 0.49 | 9.74 |
| **10.0** | 0.99 | 15.47 | 0.36 | 5.46 |

**Fig. 3 | Laser fluence per pulse (J cm⁻² pulse⁻¹) study of P2 scribe. a** SEM top-view images of P2 scribe; **b** waterfall plot of EDX spectra.

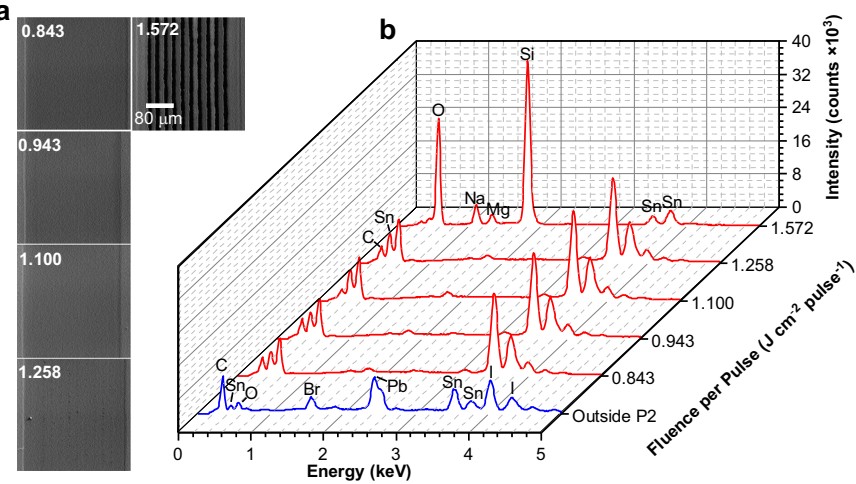

scaling further to 10 cm² led to more than a 70% reduction in PCE, along with around 34% and 53% decrease in current density and FF, respectively. These results provide a comprehensive overview on the performance of the prepared inverted PSCs, showing that both performance and reproducibility decrease significantly as the active area increases. These losses primarily stem from TCO sheet resistance and non-uniformity in spin-coated layers, which result in material defects as substrate size increases[9,15,29].

As previously mentioned, addressing the challenges of scaling up also involves optimizing the module interconnections. A crucial approach is to prepare monolithic modules composed of solar cell stripes connected in series. This design helps to compensate for the lower conductivity of the front electrode (TCO). Nevertheless, two loss mechanisms arise from these interconnections, namely: (1) contact resistance at the interconnections and (2) reduction in active area due to interconnections[24]. To maximize performance and GFF in PSMs, it is critical to minimize these individual losses by employing suitable laser scribing methods and interconnection layouts[18,22,30]. To develop a simple and industrially applicable yet highly efficient process, the laser used within this study operates at a single ablation

wavelength (here 355 nm), with a continuous repetition rate (here 30 kHz), and a single laser working distance for all three ablation processes. Additionally, laser parameters will be optimized to minimize laser heat and radiation dispersion damage to the layer stack[18,20,21]. This optimization process was conducted individually for each scribe.

Beginning with the optimization of P1 scribe, different laser fluences per pulse (the amount of energy delivered per pulse divided by the laser spot area) were applied until the conductivity between the desired isolated parts ceased, meaning that the FTO layer was fully removed. Optimal operating conditions were achieved with a single laser line and three laser passes at a fluence of 3.1 J cm⁻² pulse⁻¹ and a speed of 30 mm s⁻¹.

The goal of the P2 scribe is to create a low-ohmic resistance between the front and back electrodes. This requires the removal of several layers deposited on top of the FTO (PTAA, perovskite, PCBM, and BCP) without damaging the FTO itself. To address this challenge, techniques such as SEM and EDX analyses were employed to comprehensively investigate and qualitatively evaluate the quality of the P2 scribe. Scribing experiments with a width of 250 μm were conducted using five different laser fluences (0.843,

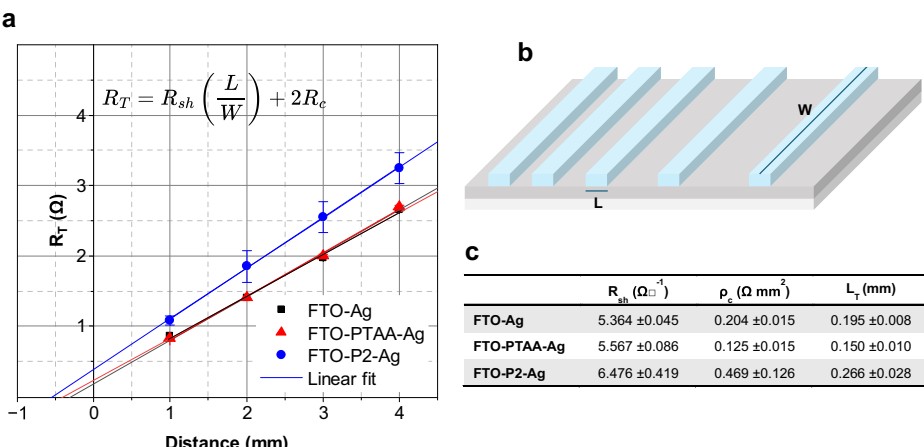

**Fig. 4 | EDX analysis in P2 and P3 scribes towards the outside scribed region. a** EDX spectra inside and outside P2 scribe; **b** SEM image and EDX elemental maps of P2; **c** EDX spectra inside and outside P3 scribe; **d** SEM image and EDX elemental maps of P3.

**Fig. 5 | Transfer length method (TLM) analysis of contacts. a** TLM plot; **b** schematic of the samples prepared for TLM tests; **c** summary table with the parameters measured in the TLM analysis.

$$R_T = R_{sh}\left(\frac{L}{W}\right) + 2R_c$$

| | $R_{sh}$ ($\Omega\square^{-1}$) | $\rho_c$ ($\Omega\,mm^2$) | $L_T$ (mm) |
|---|---|---|---|
| **FTO-Ag** | 5.364 ±0.045 | 0.204 ±0.015 | 0.195 ±0.008 |
| **FTO-PTAA-Ag** | 5.567 ±0.086 | 0.125 ±0.015 | 0.150 ±0.010 |
| **FTO-P2-Ag** | 6.476 ±0.419 | 0.469 ±0.126 | 0.266 ±0.028 |

0.943, 1.100, 1.258, and 1.572 J cm$^{-2}$ pulse$^{-1}$) as shown in Fig. 3. Data analysis revealed that the FTO layer starts to be damaged at a laser fluence of 1.572 J cm$^{-2}$ pulse$^{-1}$ or higher. Fluences between 0.843 and 1.258 J cm$^{-2}$ pulse$^{-1}$ effectively remove all material above the FTO layer, allowing for the application of the silver contact layer. Therefore, it was chosen a fluence of 0.943 J cm$^{-2}$ pulse$^{-1}$, which is an intermediate value within the fluence window. Further analysis using EDX mapping (Fig. 4) confirmed the successful removal of all active layers and the exposure of the FTO. This was evidenced by the high-intensity peaks and elemental maps of

tin and oxygen, as well as the absence of other elements from the layer stack (such as Pb, I, or Br) in the P2 scribe region. Additionally, SEM analysis of the P2 scribe edge revealed a narrow (~2 µm) region of damaged material, likely caused by laser-induced heat (Supplementary Fig. 3). Beyond this region, the morphology remains very uniform, with no observable degradation of the perovskite film.

A Transfer Length Method (TLM) analysis was performed to evaluate the contact resistance and transfer length ($L_T$) of the P2 scribed region. The results are summarized in Fig. 5. As expected, the lowest transfer length was

**Fig. 6 | Box plots of photovoltaic performance parameters for inverted 4 cm² PSCs with varying P2 scribe width and corresponding GFFs. Data from six independent batches.**

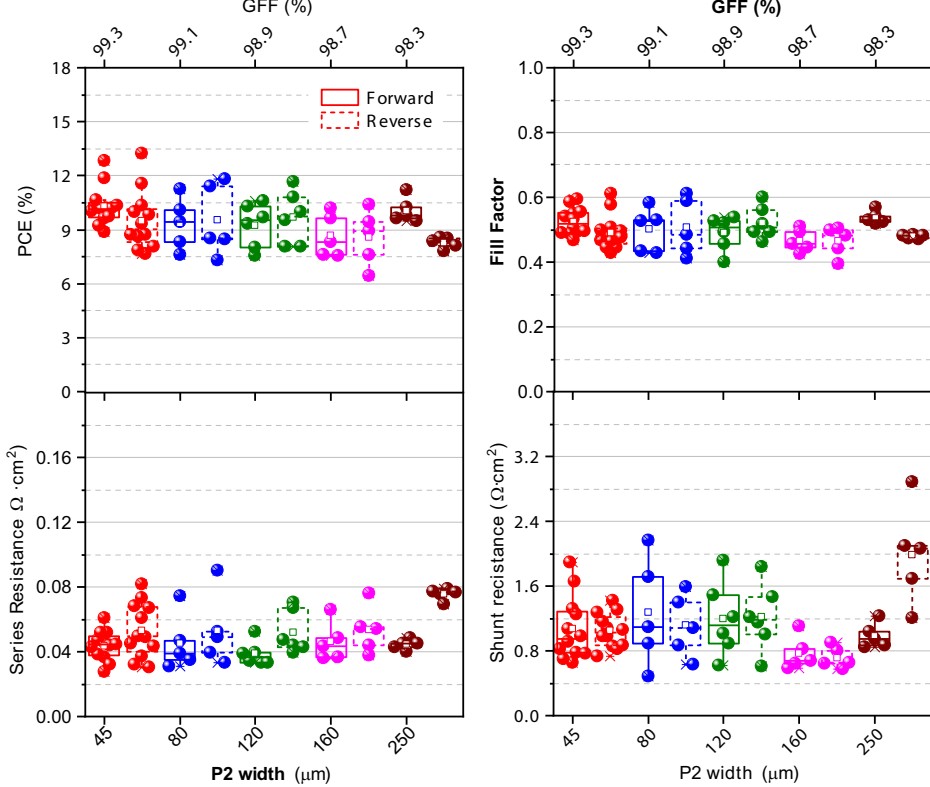

observed for the direct FTO-Ag contact. Introducing a PTAA layer between the FTO and the Ag electrode slightly reduced the transfer length from 0.195 to 0.150 mm, primarily due to a decrease in contact resistivity ($\rho_C$) from 0.204 to 0.125 $\Omega$ mm². In contrast, the P2 laser-ablated region exhibited a modest increase in transfer length to 0.266 mm, consistent with previously reported works[18].

The same laser fluence parameters used for the P2 scribe were applied to the P3 scribe, effectively removing the silver and all stacked layers up to the FTO. Once again, SEM and EDX techniques were employed (Fig. 4) to confirm the complete removal of silver within the scribe and the full exposure of the FTO.

Once the laser parameters for the different scribes were optimized, complete inverted PSMs were assembled for photovoltaic performance assessment. These inverted PSMs featured an active area of 4 cm², comprising two subcells of 2 cm² each. The width of the P1 scribe remained constant at approximately 45 μm (width of a single laser line), while P2 or P3 scribes were individually assessed for the following widths: 45, 80, 120, 140, 160, and 250 μm (see Supplementary Fig. 4). The objective is to achieve a GFF over 98%, so the distance between the scribes was kept as small as possible.

The photovoltaic parameters obtained from the J-V curves of the inverted perovskite solar modules (Figs. 6 and 7 and Supplementary Fig. 5) were used to estimate the series and shunt resistances, providing quantitative insights into the scribe quality. Series resistances are related to the ohmic contacts and are crucial for assessing the scribe quality; it was computed from the inverse of the slope of the J-V curve at the open-circuit voltage ($V_{OC}$). Minimizing series resistances is essential to maintain the device's high fill factor. On the other hand, the shunt resistance is inversely related to recombination pathways, also impacting the fill factor of the device; conversely, to series resistance, shunt resistance must be as high as possible. It was calculated from the inverse of the slope of the J-V curve near the short-circuit current density ($J_{SC}$)[20,21].

Figure 6 illustrates that varying the width of the P2 scribe has a small impact on the PSM performance when series resistances ranged from 0.025 to 0.050 $\Omega$ cm². A maximum power conversion efficiency of 13.22%, $V_{OC}$ of 2.14 V, $J_{SC}$ of 10.12 mA cm², and FF of 0.61 was achieved with both P2 and P3 scribes set at 45 μm width. However, widening the P3 scribe (while keeping P2 at 45 μm) slightly reduced both PCE and FF, followed by increased series resistances (Fig. 6). The reduction was attributed to the degradation of the active layers in the neighboring P3 scribe, caused by parasitic heating from multiple laser passes needed for wider scribes[18]. This heating led to an increase in series resistance, which directly impacted the photovoltaic performance. For all studied widths of P2 and P3, a constant shunt resistance value was observed (Figs. 6 and 7), indicating that this parameter depends solely on the quality of the layer stack, with laser scribing playing minimal to no influence on the active layer quality. Notably, the champion 4 cm² inverted PSC module was fabricated with a dead area width of ~134 μm (138 μm ± 4 μm) (Supplementary Fig. 6), resulting in an impressive GFF of 99.3%.

In addition to the devices with an active area of 4 cm², devices with 10.8 cm² with different number of subcells (5–9) were also designed and fabricated (see Fig. 8a). In this way, the optimal size and number of subcells per area could be determined, as the GFF is directly related to the number of interconnections needed. The geometry and active area of the modules were maintained, and only the width of each subcell was reduced to accommodate more subcells in the module. All the prepared modules displayed a GFF higher than 97%. Figure 9 shows the photovoltaic parameters extracted from the J-V curves as a function of the number of subcells used in the 10.8 cm² modules. The modules with 5 and 6 subcells demonstrated the best GFF and best energy conversion efficiency, respectively. As the number of cells in the modules increases, the power conversion efficiency decreases slightly, likely due to the higher number of interconnections that increases the series resistance of the module. It can also be concluded that modules with 5 subcells have lower performance because of the larger width of the subcells, which increase the ohmic losses. The corresponding fill factor

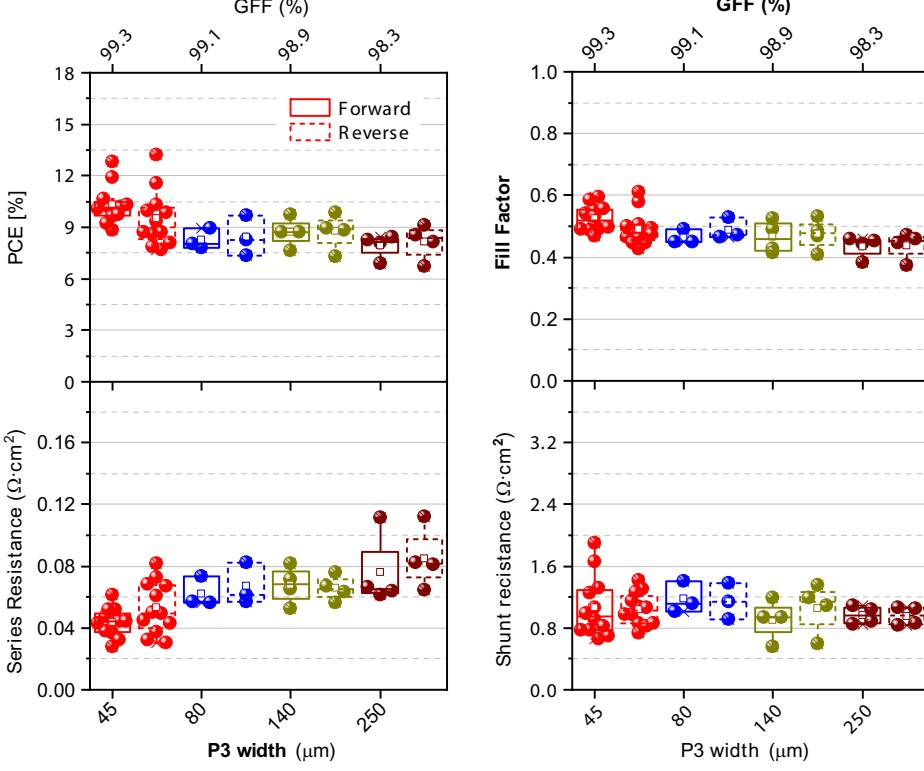

**Fig. 7 | Box plots of photovoltaic performance parameters for inverted 4 cm² PSCs with varying P3 scribe width and corresponding GFFs. Data from six independent batches.**

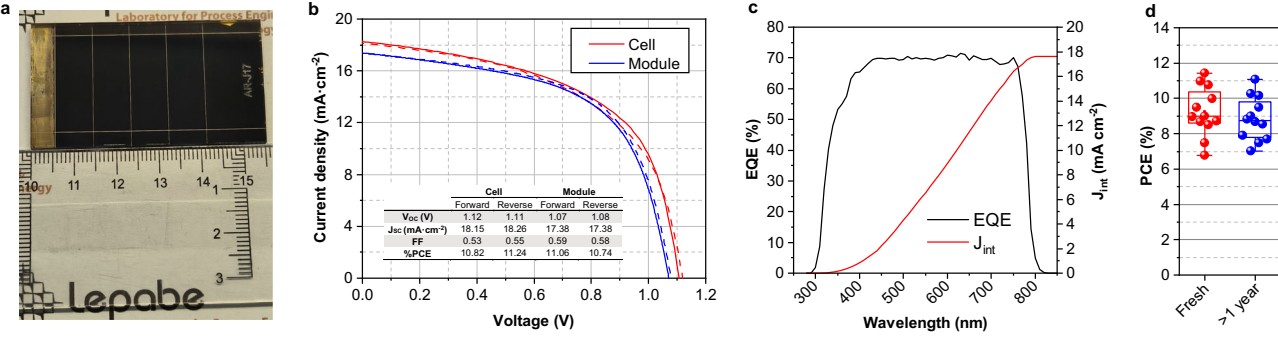

**Fig. 8 | Characterizations of inverted 10.8 cm² PSMs. a** photograph of an inverted PSM with 5 subcells (the unit of the ruler is centimeters); **b** J-V curves and photovoltaic parameters comparison of an individual PSC with a PSM with 7 subcells. **c** External quantum efficiency spectrum of a 9-subcell inverted PSM (black line) and the integrated current curve (red line). **d** Power conversion efficiency (PCE) distribution of fresh and aged (>1 year) 10.8 cm² PSC modules. The aged modules were stored in the dark under an inert atmosphere.

values support this statement, showing the same behavior as the power conversion efficiency.

Regarding the open-circuit voltage, it can be observed that each subcell in the modules generates about 1.05 V. As expected, the potential difference of the PSC module increases directly with the number of subcells since they are connected in series. Moreover, no significant variation was observed for the short-circuit current of the modules, and $J_{SC}$ is consistent with an individual cell (1.54 cm² active area)—Fig. 8b. These results lead us to the need to balance the number of cells and, consequently, increase the power output of the PSM, albeit with a decrease in the GFF. A very high GFF of 98.8% was obtained for the 5 subcell modules.

The external quantum efficiency spectrum of the 9-subcell inverted PSM shows no significant spectral losses (Fig. 8c). The integrated photocurrent, calculated from the product of the external quantum efficiency spectrum and the AM1.5G photon flux, was 17.6 mA cm⁻², in close agreement with the measured value of 17.3 mA cm⁻².

Reproducibility and long-term stability of the laser scribing process were key considerations in this study. Among the various factors affecting reproducibility, module alignment during the P2 and P3 scribing steps was identified as the primary limitation. A fabrication success rate of 60–70% was achieved (Supplementary Fig. 7), which could be improved by implementing advanced automated alignment systems. To evaluate the long-term stability of the laser-scribed interconnections, 10.8 cm² PSMs were stored in the dark under an inert atmosphere for over a year. This approach was designed to isolate any degradation effects specifically attributed to the laser processing, without interference from external stressors such as light or temperature cycling. As shown in Fig. 8d, the PCE distribution for fresh and aged modules indicates an average PCE loss of less than 5%, suggesting that laser scribing does not lead to significant long-term degradation.

## Conclusions
In this work, it has been demonstrated the feasibility of producing a continuous P1-P2-P3 interconnection using a single nanosecond laser

**Fig. 9 |** Box plots of photovoltaic parameters of inverted 10.8 cm² PSMs with different numbers of subcells. Note that n corresponds to the number of cells that compose the module, and the current density was normalized by the subcell area. The values represent data from 4 independent batches.

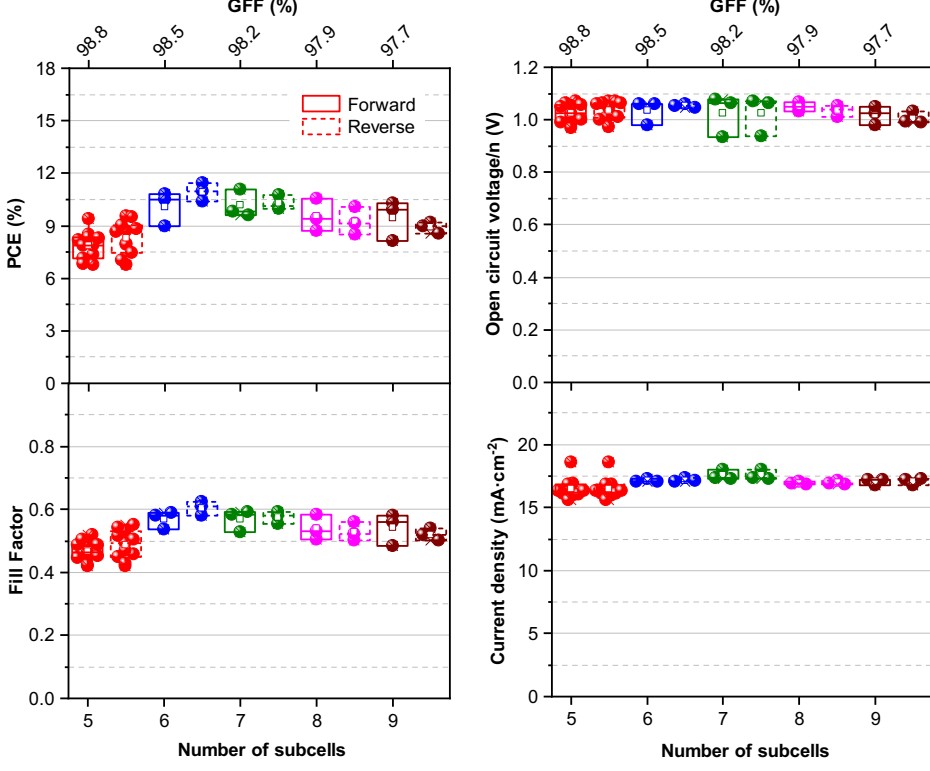

source for all three scribes. By optimizing the laser parameters, the dead area in the inverted perovskite solar module was significantly reduced to 0.7% for a 4 cm² active area and 1.2% for a 10.8 cm² active area. These optimized laser conditions were applied to make p-i-n PSC modules with exceptionally high geometrical fill factors. An extensive investigation into the optimal size and number of subcells in modules with a 10.8 cm² active area revealed that configurations with 6 or 7 subcells achieved the best performance, with GFF values above 98%. However, a further increase in the number of subcells slightly reduced the performance of the module due to an increase in the series resistance of the interconnections. The interconnection width of ~134 µm enabled inverted PSMs with a GFF of 99.3% for a 4 cm² active area module with 2 subcells and 98.8% for a 10.8 cm² active area module with 5 subcells. To the best of the authors' knowledge, these represent the highest GFFs reported for a continuous P1-P2-P3 scribing process, marking an important advancement in the field.

## Data availability

The data that support the findings of this study are available from the corresponding author upon reasonable request.

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

## Acknowledgements

A.E.R.S. acknowledges the Portuguese Foundation for Science and Technology (FCT) through the Eugloh Alliance for the PhD grant (PRT/BD/154972/2023). This work is financially supported by: national funds through the FCT/MCTES (PIDDAC), under the project PTDC/EQU-EQU/4193/2021 with https://doi.org/10.54499/PTDC/EQU-EQU/4193/2021 (https://doi.org/10.54499/PTDC/EQU-EQU/4193/2021); national funds through FCT/MCTES (PIDDAC): LEPABE, UIDB/00511/2020 (https://doi.org/10.54499/UIDB/00511/2020) and UIDP/00511/2020 (https://doi.org/10.54499/UIDP/00511/2020) and ALiCE, LA/P/0045/2020 (https://doi.org/10.54499/LA/P/0045/2020). V.C.M.D. thanks the support of Agenda "H2Driven Green Agenda", nr. C644923817-00000037, investment project nr. 50, financed by the Recovery and Resilience Plan (PRR) and by the European Union—NextGeneration EU.

## Author contributions

A.S. contributed to the fabrication of PSC cells and modules, characterization, laser scribing optimization, modules design, and drafting the original manuscript. V.C.M.D. contributed to module design, SEM and EDX analysis, methodology, supervision, and both writing and reviewing the manuscript. A.M. contributed to reviewing the manuscript and funding acquisition. L.A. contributed to conceptualization and methodology, funding acquisition, project administration, supervision, and reviewing the manuscript.

## Competing interests

The authors declare no competing interests.
