## [Transparent Peer Review file · Communications Engineering]

Inverted perovskite solar modules with 99.3% geometrical fill factor via nanosecond single laser patterning.

Corresponding Author: Dr Luísa Andrade

Version 0:

Reviewer comments:

Reviewer #1

(Remarks to the Author)

Recommendation: reject (as noted below)

The manuscript from Soto et al. demonstrated a high geometrical fill factor (GFF) of 99.3% in inverted perovskite solar modules (PSMs) using a single nanosecond UV laser. By optimizing laser parameters, the dead zone was minimized to 0.7%, reducing interconnection losses and enhancing module efficiency.

However, the figures in the manuscript appears insufficient to fully support the optimization of laser scribing conditions, and the technical justification for the laser processing lacks clarity. Additionally, the module efficiency is relatively low compared to previous published results, and the images depicting the laser-processed areas are not sufficiently clear to experimental findings. Therefore, I do not find this manuscript suitable for publication in this journal.

Comments:

1. How was it demonstrated that no thermal damage occurred during the P1-P2-P3 process using a single 355 nm nanosecond laser? Has a thorough evaluation been performed to investigate potential device damage, with dead zones being significantly reduced? Are the SEM and EDX analysis results sufficient evidence to support this claim?
2. Compared to previous studies using picosecond and femtosecond lasers, what are the advantages and disadvantages of using a nanosecond laser?
3. How was it verified that the increase in the geometrical fill factor (GFF) directly led to an improvement in power conversion efficiency (PCE)? Was a clear comparison made between the electrical performance of the 99.3% GFF module and previous modules with GFFs ranging from 90% to 96%?

Reviewer #2

(Remarks to the Author)

This manuscript presents a technological advancement in the fabrication of inverted perovskite solar modules (PSMs), demonstrating an optimized laser scribing strategy that enables high geometric fill factors (GFF) using a single nanosecond laser source. The authors successfully achieve GFF values of 99.3% for 4 cm² active area modules and 98.8% for 10.8 cm² modules, surpassing some of the previously reported values in the literature. This contribution is particularly noteworthy as it utilizes an industrial-friendly nanosecond laser, making the process more scalable compared to previous works relying on picosecond or femtosecond lasers.

The study is well-structured, with a clear experimental methodology, systematic optimization of laser scribing parameters, and comprehensive characterization of the fabricated modules. The use of SEM-EDX analysis to verify material removal during laser scribing is a valuable addition to ensuring interconnection quality.

Despite these strengths, there are several areas where the study should be improved to provide a more comprehensive understanding of the device performance. A notable limitation is the significant efficiency loss observed when scaling up from small-area cells (max PCE 18.80%) to modules (max PCE 13.22% for 4cm² and even lower for larger modules). While the authors acknowledge the role of increased series resistance and interconnection losses, additional analysis, such as Transmission Line Method (TLM) measurements, could help quantify contact resistance and provide deeper insights into interfacial losses at the laser-scribed regions. This would strengthen the claim that the proposed laser scribing approach minimizes losses without introducing significant recombination or resistance effects. I am expecting a max reduction of the efficiency going from small area to 4cm² of around 10%.

The long-term stability of the modules is not discussed, which is crucial for assessing the viability of this approach where the total energy transfer is generally larger than the ps lasers. Since laser processing can enhance PSC degradation during accelerated test, it would be beneficial to include accelerated aging tests (e.g., light-soaking, thermal cycling) to determine the robustness of the scribed interconnections over time.

Another aspect that could enhance the study is the inclusion of external quantum efficiency (EQE) measurements. While PCE is characterized through J-V measurements, EQE would provide valuable insights into potential spectral losses and the effectiveness of charge collection across different wavelengths and confirming the measured JSC.

Finally, the manuscript does not provide explicit information on fabrication success rates or yield statistics. Given that industrial scalability is a key motivation for this work, reporting the percentage of successfully fabricated modules and their performance variation across multiple batches would provide additional confidence in the reproducibility of the results.

Overall, this study represents a meaningful step forward in the development of scalable perovskite solar modules with high interconnection efficiency. The proposed laser scribing strategy is an industrial-friendly approach that significantly reduces dead area losses, making it highly relevant for future commercialization efforts. However, addressing the outlined concerns—particularly efficiency scaling, interfacial resistance analysis, long-term stability, and additional optical/electrical characterizations—would strengthen the manuscript and provide a more holistic assessment of the proposed methodology. I encourage the authors to address these aspects to further enhance the impact of their work.

Reviewer #3

(Remarks to the Author)

Dear Author,

I've been reading your article about your work with much pleasure and interest.

And I was glad you also used the discontinuous point contact of Rakocevic (25) and Di Giacomo (26) in the overview. As the 99.6% efficiency of Di Giacomo is higher than yours, it is a more complex scribing method with regard to layout and alignment of the scribes.

I do agree with you that a nanosecond laser would be preferred for industrial use. But the prices of pico lasers are dropping. And a third harmonic could be in the end, be more expensive in use than a fundamental (1R) pico second laser. As the optics are more expensive and the electrical efficiency is less.

The used UV laser was described in the article, but I do miss a value for the spotsize (focussize) and focal length of the optics used. I could assume that it will be close to 45 micron, stated as a single laser line, but that is a guess.

The method/strategy to obtain larger scribe widths is not mentioned, only the resulting scribe width of (45,80,120,140,160,250 micron). In the Figure 4a I see some much smaller lines (right side) but they look a lot smaller than the 45 micron of a single laser line.

Not only the cell area but also the cell width is important with regard to efficiency and GFF. A picture of the layout of the 2 cell test module would help a lot. A wider cell width automatically has a better GFF as the dead zone stays the same.

On page 8, you use focus length. I would prefer the term focal length

Another issue is the Pb content of the perovskite. As no mention was made to safety. I do hope that you use a sufficient method to remove the ablation debris, in order to avoid any contamination and toxication of operators. One 100 mm P2 scribe (20 micron) line equals the daily allowed occupational exposure limit (OEL)

Kind Regards, Johan Bosman

Version 1:

Reviewer comments:

Reviewer #1

(Remarks to the Author)

This review addresses the key points requested. With this revision, the manuscript is now ready for publication.

Reviewer #2

(Remarks to the Author)

The Authors addressed all my requests with additional experiments and descriptions. The manuscript can be published as it is.

Reviewer #3

(Remarks to the Author)

I have a question about the laser optics in the part Module Fabrication (page 6-7).

What do you mean with: "A laser optics with 10 cm was used".

How did you determined the optimal focal length in "optimal focal length of 434 mm was determined" (part Module Fabrication, page 7).

Dear Editor and Reviewers,

We appreciate the opportunity to revise our manuscript entitled "*Inverted perovskite solar modules with 99.3% geometrical fill factor via nanosecond single laser patterning*". We are grateful for the insightful comments provided by the reviewers, which have been highly valuable in improving the quality of our work. All revisions and additions to the manuscript are highlighted in green. We have made every effort to address all reviewer comments and suggestions, and we hope that the revised manuscript now meets the criteria for publication in *Communications Engineering*.

Reviewer #1 (Remarks to the Author):

Recommendation: reject (as noted below)

The manuscript from Soto et al. demonstrated a high geometrical fill factor (GFF) of 99.3% in inverted perovskite solar modules (PSMs) using a single nanosecond UV laser. By optimizing laser parameters, the dead zone was minimized to 0.7%, μ processing lacks clarity. Additionally, the module efficiency is relatively low compared to previous published results, and the images depicting the laser-processed areas are not sufficiently clear to experimental findings. Therefore, I do not find this manuscript suitable for publication in this journal.

Reviewer Comment:

1. How was it demonstrated that no thermal damage occurred during the P1-P2-P3 process using a single 355 nm nanosecond laser? Has a thorough evaluation been performed to investigate potential device damage, with dead zones being significantly reduced? Are the SEM and EDX analysis results sufficient evidence to support this claim?

Answer:

The authors agree with the reviewer's concern. In this work, an extensive optimization of the laser scribing parameters was conducted. A 355 nm ultraviolet wavelength laser was chosen to minimize residual heat effects on the areas surrounding the scribes. To assess these effects, both SEM and EDS analyses were performed throughout the scribing optimization process to ensure high scribing quality, uniformity, and minimal damage to the adjacent regions. SEM analysis provided detailed insights into the morphology of the scribes, while EDS confirmed effective material removal and the preservation of surrounding areas by detecting key elemental distribution.

SEM analysis of the P2 scribe edge revealed a narrow region of damaged material, likely caused by laser-induced heat. As shown in Figure S2, a $\sim 2 \mu\text{m}$

border of damaged material was observed, which is an acceptable extent considering the total scribe width of 45 μm). Beyond this region, the morphology remains uniform, with no signs of degradation in the perovskite film layer.

Figure S2. SEM image of the P2 scribe border, showing the damaged material caused by the laser pass.

In the manuscript was included: Additionally, SEM analysis of the P2 scribe edge revealed a narrow ($\sim 2 \mu\text{m}$) region of damaged material, likely caused by laser-induced heat (Figure S2). Beyond this region, the morphology remains very uniform, with no observable degradation of the perovskite film.

Reviewer Comment:

2. Compared to previous studies using picosecond and femtosecond lasers, what are the advantages and disadvantages of using a nanosecond laser?

Answer:

Our research group is focused on transitioning the preparation of lab-scale devices to large-area applications. Accordingly, we are committed to developing scalable fabrication methods for perovskite solar cells and modules that meet industrial manufacturing requirements. In this context, nanosecond lasers offer a particularly attractive solution due to their high energy efficiency and significantly lower cost compared to femtosecond (fs) and picosecond (ps) lasers, which are often prohibitively expensive for large-scale production. While nanosecond lasers do present certain drawbacks, such as greater thermal effects, heat diffusion, and higher energy per pulse compared to fs/ps lasers, our work demonstrates that they can be effectively employed for ablating inverted perovskite solar modules. This approach offers a viable and cost-efficient pathway for scaling up fabrication processes without compromising device performance.

Reviewer Comment:

3. How was it verified that the increase in the geometrical fill factor (GFF) directly led to an improvement in power conversion efficiency (PCE)? Was a clear comparison made between the electrical performance of the 99.3% GFF module and previous modules with GFFs ranging from 90% to 96%?

Answer:

We thank the reviewer for this important observation. The dead area in a module, primarily consisting of the interconnection regions, does not contribute to power generation and thus negatively impacts overall power output. In conventional modules with a GFF of around 90%, a typical power loss of approximately 10% can occur due to these inactive regions. In our study, a direct comparison was made between modules with different GFF values to evaluate the impact on power conversion efficiency (PCE). Specifically, for the 2-subcell modules, we maintained a constant active area of 4 cm² (each subcell measuring 2×1 cm²) while systematically varying the scribe widths to tune the GFF, thereby isolating the effect of GFF on device performance. As shown in Figures 6 and 7, an increase in GFF, from values typical of 90–96% up to 99.3%, resulted in a clear improvement in PCE. This enhancement is attributed not only to the reduction of the dead area, as described by the relation $GFF = \text{active area} / (\text{active area} + \text{dead area}) \times 100$, but also to improved electrical performance due to reduced series resistance at the P2 and P3 interconnections.

For the larger-area modules (10.8 cm² active area), we maintained a constant subcell length while reducing the subcell width and increasing the number of interconnections. As expected, this led to a decrease in GFF with increasing numbers of subcells (since GFF decreases as the number of interconnections increases, following the $n-1$ relationship where n is the number of subcells), and a corresponding decrease in PCE. An exception was observed in the 5-subcell module: although it exhibited a higher GFF, it showed a lower PCE. This was due to the increased subcell width, which introduced higher ohmic losses and illustrates that optimal cell geometry and resistance management must also be considered alongside GFF.

In conclusion, our results show a clear correlation between increased GFF and improved PCE when other parameters are controlled. This emphasizes the importance of minimizing dead area through precise, high-quality scribing to reduce power losses and improve module efficiency during scale-up.

Reviewer #2 (Remarks to the Author):

This manuscript presents a technological advancement in the fabrication of inverted perovskite solar modules (PSMs), demonstrating an optimized laser scribing strategy that enables high geometric fill factors (GFF) using a single nanosecond laser source. The authors successfully achieve GFF values of 99.3% for 4 cm² active area modules and 98.8% for 10.8 cm² modules, surpassing some

of the previously reported values in the literature. This contribution is particularly noteworthy as it utilizes an industrial-friendly nanosecond laser, making the process more scalable compared to previous works relying on picosecond or femtosecond lasers.

The study is well-structured, with a clear experimental methodology, systematic optimization of laser scribing parameters, and comprehensive characterization of the fabricated modules. The use of SEM-EDX analysis to verify material removal during laser scribing is a valuable addition to ensuring interconnection quality. Despite these strengths, there are several areas where the study should be improved to provide a more comprehensive understanding of the device performance.

Reviewer comment:

A notable limitation is the significant efficiency loss observed when scaling up from small-area cells (max PCE 18.80%) to modules (max PCE 13.22% for 4cm² and even lower for larger modules). While the authors acknowledge the role of increased series resistance and interconnection losses, additional analysis, such as Transmission Line Method (TLM) measurements, could help quantify contact resistance and provide deeper insights into interfacial losses at the laser-scribed regions. This would strengthen the claim that the proposed laser scribing approach minimizes losses without introducing significant recombination or resistance effects. I am expecting a max reduction of the efficiency going from small area to 4cm² of around 10%.

Answer:

The authors acknowledge the reviewer's comment and have conducted a Transfer Length Method (TLM) analysis to evaluate the quality of the P2 scribing achieved using the optimized laser parameters. As expected, the lowest transfer length (L_T) was observed for the FTO-Ag contact, in agreement with values reported in the literature.[ref 18 manuscript] A detailed discussion of the TLM results, along with a summary table and corresponding plot, has been added to the manuscript.

In the manuscript was included: A Transfer Length Method (TLM) analysis was performed to evaluate the contact resistance and transfer length (L_T) of the P2 scribed region. The results are summarized in Figure 5. As expected, the lowest transfer length was observed for the direct FTO-Ag contact. Introducing a PTAA layer between the FTO and the Ag electrode slightly reduced the transfer length from 0.195 to 0.150 mm, primarily due to a decrease in contact resistivity (ρ_c) from 0.204 to 0.125 Ω mm². In contrast, the P2 laser-ablated region exhibited a

modest increase in transfer length to 0.266 mm, consistent with previously reported works.¹⁸

Figure 5: Transfer length method (TLM) analysis of contacts: a) TLM plot; b) schematic of the samples prepared for TLM tests; c) summary table with the parameters measured in the TLM analysis.

Reviewer comment:

The long-term stability of the modules is not discussed, which is crucial for assessing the viability of this approach where the total energy transfer is generally larger than the ps lasers. Since laser processing can enhance PSC degradation during accelerated test, it would be beneficial to include accelerated aging tests (e.g., light-soaking, thermal cycling) to determine the robustness of the scribed interconnections over time.

Answer:

The authors acknowledge the reviewers' point. Accelerated aging tests were not performed in this study, as such tests typically involve light and thermal cycling, which can obscure specific degradation effects associated with laser processing. To isolate potential degradation induced by laser scribing, 10.8 cm² modules with multiple laser-scribed interconnections were stored in the dark under an inert atmosphere for over a year. Under these controlled conditions, an average performance loss of less than 5 % was observed, indicating that laser scribing does not lead to significant degradation over time.

In the manuscript was included: To evaluate the long-term stability of the laser-scribed interconnections, 10.8 cm² PSMs were stored in the dark under an inert atmosphere for over a year. This approach was designed to isolate any degradation effects specifically attributed to the laser processing, without interference from external stressors such as light or temperature cycling. As shown in Figure 10d, the PCE distribution for fresh and aged modules indicates an average PCE loss of less than 5 %, suggesting that laser scribing does not lead to significant long-term degradation.

Figure 10d. Power conversion efficiency (PCE) distribution of fresh and aged (>1 year) 10.8 cm² PSC modules. The aged modules were stored in the dark under an inert atmosphere.

Reviewer comment:

Another aspect that could enhance the study is the inclusion of external quantum efficiency (EQE) measurements. While PCE is characterized through J-V measurements, EQE would provide valuable insights into potential spectral losses and the effectiveness of charge collection across different wavelengths and confirming the measured JSC.

Answer:

The authors acknowledge the reviewers' comments. The IPCE of a 9-subcell module was measured, and the resulting spectrum is presented in Figure 10c (black line). The integrated product of the EQE spectrum and the AM1.5G photon flux are also shown (red line). The integrated current of 17.6 mA·cm⁻² closely matches the measured value of 17.3 mA·cm⁻², indicating that no spectral losses were observed.

In the manuscript was included: The EQE spectrum of the 9 subcell inverted PSM shows no significant spectral losses (Figure 10c). The integrated photocurrent, calculated from the product of the EQE spectrum and the AM1.5G photon flux, was 17.6 mA·cm⁻², in close agreement with the measured value of 17.3 mA·cm⁻².

Figure 10c. EQE spectrum of a 9 subcell inverted PSM (black line) and the integrated current curve (red line).

Reviewer comment:

Finally, the manuscript does not provide explicit information on fabrication success rates or yield statistics. Given that industrial scalability is a key motivation for this work, reporting the percentage of successfully fabricated modules and their performance variation across multiple batches would provide additional confidence in the reproducibility of the results.

Answer:

The authors appreciate the reviewer's comment. To ensure reproducibility of the results, multiple independent batches of PSC modules were fabricated in this study. All performance data presented reflect results from several batches: specifically, 6 batches of 12 modules each for 4 cm² active area and 4 batches of 8 modules each for 10.8 cm² active area. A success rate of 60-70 % was achieved during module fabrication, with module alignment during the laser scribing step identified as the primary limiting factor (see Figure S5). Nevertheless, the alignment issue is specific to the laboratory setting, as the laser system is manually aligned to maintain versatility for other applications in our host laboratory. This limitation is easily addressed at the industrial scale, where automated alignment systems are standard.

In the manuscript was included: Among the various factors affecting reproducibility, module alignment during the P2 and P3 scribing steps was identified as the primary limitation. A fabrication success rate of 60-70 % was achieved (Figure S5), which could be significantly improved by implementing advanced automated alignment systems.

Figure S5. Fabrication success rates for prepared PSC modules.

Reviewer comment:

Overall, this study represents a meaningful step forward in the development of scalable perovskite solar modules with high interconnection efficiency. The proposed laser scribing strategy is an industrial-friendly approach that significantly reduces dead area losses, making it highly relevant for future commercialization efforts. However, addressing the outlined concerns—particularly efficiency scaling, interfacial resistance analysis, long-term stability, and additional optical/electrical characterizations—would strengthen the manuscript and provide a more holistic assessment of the proposed methodology.

I encourage the authors to address these aspects to further enhance the impact of their work.

Reviewer #3 (Remarks to the Author):

Dear Author, I've been reading your article about your work with much pleasure and interest. And I was glad you also used the discontinuous point contact of Rakocevic (25) and Di Giacomo (26) in the overview. As the 99.6% efficiency of Di Giacomo is higher than yours, it is a more complex scribing method with regard to layout and alignment of the scribes.

Reviewer Comment:

I do agree with you that a nanosecond laser would be preferred for industrial use. But the prices of pico lasers are dropping. And a third harmonic could be in the end, be more expensive in use than a fundamental (IR) pico second laser. As the optics are more expensive and the electrical efficiency is less.

Answer:

The authors acknowledge and agree with the reviewer's safeguard. Nevertheless, while picosecond (ps) lasers are becoming more affordable, nanosecond (ns) lasers still remain the more practical and cost-effective choice for industrial applications. They are widely available, robust, and significantly cheaper in terms of both capital and maintenance costs. In high-throughput manufacturing, such as for perovskite solar modules, the precision of ps lasers is often unnecessary. Optimized ns laser processing can achieve excellent scribing quality with minimal thermal damage, as demonstrated in our work. Additionally, UV ns lasers offer better material absorption for perovskite and TCO layers, enabling clean ablation despite slightly lower efficiency. Overall, ns lasers provide the best balance of performance, scalability, and cost for industrial-scale deployment.

Reviewer comment:

The used UV laser was described in the article, but I do miss a value for the spot size (focussize) and focal length of the optics used. I could assume that it will be close to 45 micron, stated as a single laser line, but that is a guess. The method/strategy to obtain larger scribe widths is not mentioned, only the resulting scribe width of (45,80,120,140,160,250 micron). In the Figure 4a I see some much smaller lines (right side) but they look a lot smaller than the 45 micron of a single laser line.

Answer:

The scribing optimization process was conducted using laser optics with 10 cm, and an optimal focal length of 434 mm was determined. At this focal length, an average spot size of 45 μm was achieved when drawing a single line with the laser system, resulting in a scribe line width of approximately 45 μm . For wider scribes, a rectangular pattern was used instead of a single line. In these cases, a parallel line fill within the rectangle was implemented, and the spacing between lines was carefully optimized. This spacing was adjusted to minimize significant overlap between adjacent laser passes, which could otherwise lead to deeper penetration and undesirable material removal. A line spacing of 0.040 mm was found to provide good uniformity across the scribe area, while effectively minimizing overlap.

Figure 4a displays images of a 250 μm wide scribe produced using the same line spacing but varying laser fluences. The final image on the right demonstrates that higher laser fluence leads to increased penetration in the overlaps regions, causing removal of the FTO material and exposing the underlying glass substrate (visible as dark areas).

Reviewer comment:

Not only the cell area but also the cell width is important with regard to efficiency and GFF. A picture of the layout of the 2 cell test module would help a lot. A wider cell width automatically has a better GFF as the dead zone stays the same.

Answer:

The authors acknowledge the reviewer's comment and have included a layout scheme for the 2 subcell module (Figure S3). As shown, the active area of 4 cm^2 (with individual cells measuring 2 cm (length) \times 1 cm (width)) was maintained, and only the width of the P2 and P3 scribes was adjusted individually to assess their impact on the power conversion efficiency.

Figure S3: Layout of a 2 subcell PSC module showing the enlargement of the P3 scribe (the same strategy was applied to P2 scribe).

Reviewer comment: On page 8, you use focus length. I would prefer the term focal length

Answer:

The authors acknowledge the reviewer's comment and have changed the term “focus length” to “focal length” as suggested.

Reviewer comment:

Another issue is the Pb content of the perovskite. As no mention was made to safety. I do hope that you use a sufficient method to remove the ablation debris, in order to avoid any contamination and toxication of operators. One 100 mm P2 scribe (20 micron) line equals the daily allowed occupational exposure limit (OEL)

Answer:

The authors acknowledge the reviewer's comment. The presence of lead is indeed a significant concern. To address this, our laboratory's laser ablation system was specially designed with safety in mind. The UV laser is fully enclosed within a dedicated chamber, and the access door remains closed during operation. This setup not only prevents exposure to laser radiation but also contains any debris or dust generated during materials ablation, ensuring the operator's safety. Additionally, an internal exhaust system was installed to effectively extract and filter all debris, further minimizing the risk of lead contamination.

Dear Editor and Reviewers,

We appreciate the opportunity to revise our paper one final time in response to the reviewers' remaining concerns. Below, we provide a detailed, point-by-point response to each comment. All revisions and additions made to the manuscript are highlighted in green. We have also ensured that the manuscript adheres fully to the formatting and policy requirements of *Communications Engineering* journal.

Reviewer #3:

Reviewer comment: I have a question about the laser optics in the part Module Fabrication (page 6-7). What do you mean with: "A laser optics with 10 cm was used".

Answer:

We appreciate the reviewer for pointing out this important observation. We have clarified the terminology in our procedure. Specifically, we used an F-theta scan lens with a focal length of 160 mm, and we determined an optimal working distance of 200 mm.

The manuscript was updated as follows: "A F-theta scan lens with a focal length of 160 mm was used, and an optimal working distance of 200 mm was determined."

Reviewer comment: How did you determined the optimal focal length in "optimal focal length of 434 mm was determined" (part Module Fabrication, page 7).

Answer:

To determine the optimal working distance (200 mm), we adjusted the distance between the laser lens and the substrate until reaching the minimum spot size where the laser causes minimal damage to the layer stack. This optimal spot size was 45 μm . At this distance, an excellent reproducibility was observed in the scribe widths obtained.

Dear Editor and Reviewers,

We appreciate the opportunity to revise our paper one final time in response to the reviewers' remaining concerns. Below, we provide a detailed, point-by-point response to each comment. All revisions and additions made to the manuscript are highlighted in green. We have also ensured that the manuscript adheres fully to the formatting and policy requirements of *Communications Engineering* journal.

Reviewer #3:

Reviewer comment: I have a question about the laser optics in the part Module Fabrication (page 6-7). What do you mean with: "A laser optics with 10 cm was used".

Answer:

We appreciate the reviewer for pointing out this important observation. We have clarified the terminology in our procedure. Specifically, we used an F-theta scan lens with a focal length of 160 mm, and we determined an optimal working distance of 200 mm.

The manuscript was updated as follows: "A F-theta scan lens with a focal length of 160 mm was used, and an optimal working distance of 200 mm was determined."

Reviewer comment: How did you determined the optimal focal length in "optimal focal length of 434 mm was determined" (part Module Fabrication, page 7).

Answer:

To determine the optimal working distance (200 mm), we adjusted the distance between the laser lens and the substrate until reaching the minimum spot size where the laser causes minimal damage to the layer stack. This optimal spot size was 45 μm . At this distance, an excellent reproducibility was observed in the scribe widths obtained.